# Simple and Fast CNN for Vision

## Abstract

Traditional Convolutional Neural Networks (CNNs) tend to use $3 \times 3$ small kernels, but can only capture limited neighboring spatial information. Inspired by the success of Vision Transformers (ViTs) in capturing long-range visual dependencies, recent CNNs have reached a consensus on utilizing large kernel convolutions (e.g., astonishingly, 111 kernel). Nevertheless, these approaches are unfriendly to hardware, imposing a serious computation burden on training or inference. This paper introduces a Simple and Fast Convolutional Neural Network (SFCNN) that employs a sequence of stacked $3 \times 3$ convolutions but surpasses state-of-the-art CNNs with larger kernels. In particular, we build a thin and deep model, which encourages more $3 \times 3$ convolutions to capture more spatial information under the limited computing complexity rather than opting for a heavier and shallower architecture. To further enlarge the receptive field, we redesign the traditional inverted residual bottleneck with two $3 \times 3$ depthwise convolutions. In addition, we propose a novel Global Sigmoid Linear Unit (GSiLU) activation function to capture global coarse-grained spatial information. Our SFCNN performs better than state-of-the-art CNNs and ViTs on various tasks, including ImageNet-1K image classification, COCO instance segmentation, and ADE20K semantic segmentation. It also has good scalability and outperforms existing state-of-the-art lightweight models. All materials containing codes and logs have been included in the supplementary materials.

## 1 Introduction

Neural network architecture holds paramount significance in machine learning and computer vision research. In recent years, notable Vision Transformer (ViT) (Dosovitskiy et al., 2021; Touvron et al., 2021) with global attention have considerably enhanced the performance of various computer vision tasks and surpassed convolutional neural networks (CNNs) by a large margin.

Recently, the Swin Transformer (Liu et al., 2021) proposes local shift-window attention and obtains better results than ViTs (Dosovitskiy et al., 2021) with the global window. This local attention is viewed as a variant of the large kernel. Thus, some novel CNNs use large convolutional kernels to compete with ViTs. Both DWNet (Han et al., 2022) and ConvNeXt (Liu et al., 2022) obtain better results by replacing the local attention in Swin (Liu et al., 2021) with the $7 \times 7$ depthwise convolution (DWConv). Following this large kernel design, Table 1 shows many CNN-based architectures, and the largest kernel size is even 111. In addition, as shown in Figure 1, some large kernel methods (Ding et al., 2024; Xu et al., 2023; Li et al., 2024; Yu et al., 2024) are unfriendly to hardware, thus increasing the difficulty and complexity in the training and inference stages.

Is the large kernel CNN needed? Previous small-kernel CNNs (He et al., 2016; Xie et al., 2017; Sandler et al., 2018; Radosavovic et al., 2020) focus more on designing new bottlenecks and ignoring the importance of the receptive field; therefore, they cannot model long-range dependencies and obtain unsatisfactory results. This paper stacks $3 \times 3$ DWConvs in a simple CNN architecture and outperforms state-of-the-art CNNs and ViTs (efficiency and effectiveness). In particular, we make some simple but effective designs to let $3 \times 3$ convolutions progressively capture various sizes of visual cues in one block, which breaks through the limitation of small kernels. First, we design a thin and deep model to capture more spatial information instead of a heavy and shallow one, which could have more $3 \times 3$ convolutions under the same computing complexity. We then redesign the traditional inverted residual bottleneck (Sandler et al., 2018) with two $3 \times 3$ DWConvs, to further enlarge the receptive field. Finally, we replace the input of the popular Sigmoid Linear

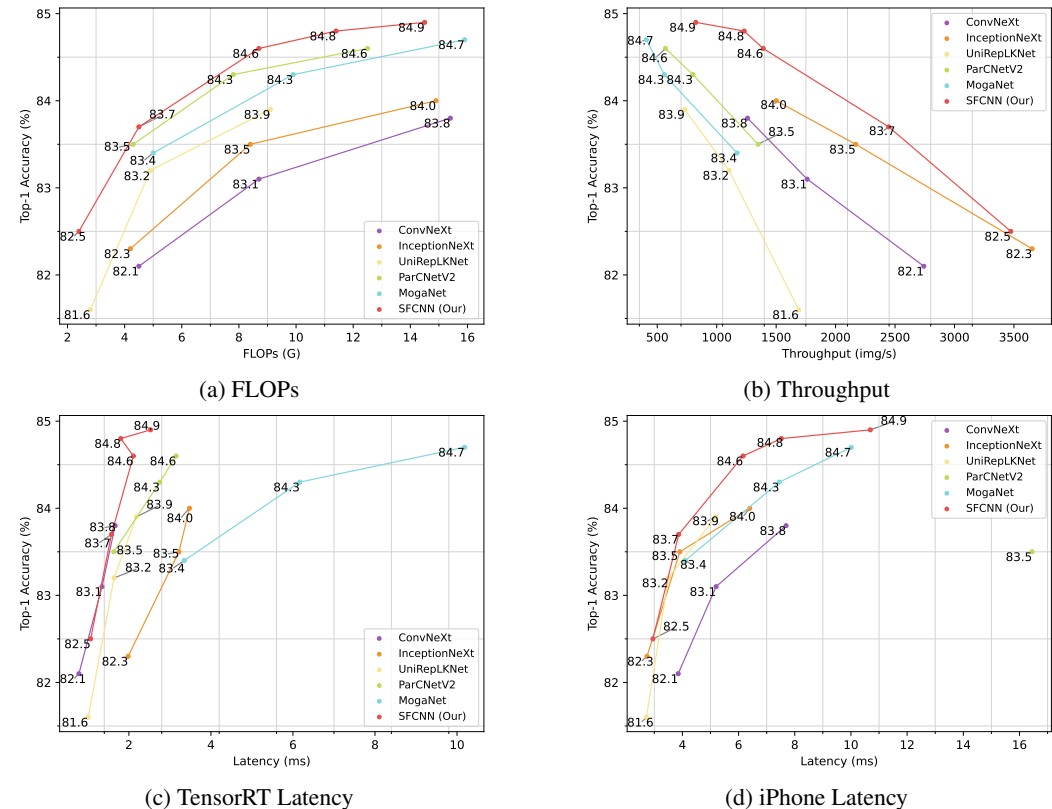

Figure 1: Comparing the accuracy with FLOPs (a), Throughput (b), TensorRT Latency (c), and iPhone Latency (d) with ConvNeXt (Liu et al., 2022), InceptionNeXt (Yu et al., 2024), UniRepLKNet (Ding et al., 2024), ParCNetV2 (Xu et al., 2023) and MogaNet (Li et al., 2024) on ImageNet-1K. Throughput is tested on a Nvidia 4090 GPU with PyTorch. TensorRT latency is tested on a 4090 GPU with TensorRT, and iPhone latency is tested on an iPhone SE3 with Core ML. Figure (d) only marks one result of ParCNetV2, because two larger versions cost more than 100ms.

Unit (SiLU) activation function with global average pooled features to capture global coarse-grained spatial information. Impressively, the overall SFCNN architecture is simple and fast and outperforms existing complicated architectures.

Figure 1 shows that our SFCNN achieves the best accuracy in ImageNet-1K image classification under four computational complexity measures, compared to other large-kernel CNNs. ConvNeXt (Liu et al., 2022) is the pioneer in this field but only performs well on TensorRT latency. Inception-NeXt (Yu et al., 2024) enjoys high throughput and iPhone latency, but FLOPs and TensorRT latency are unsatisfactory. UniRepLKNet (Ding et al., 2024) uses the re-parameterization technique; thus, it performs well on iPhone latency but shows poor results on FLOPs and throughput, and its performance on TensorRT is also bad. ParCNetV2 (Xu et al., 2023) introduces huge kernel sizes (even 111), and MogaNet (Li et al., 2024) introduces a gate mechanism. Both of the above techniques are unfriendly to hardware devices. Thus, they are terrible on real-world measures (throughput, TensorRT latency, and iPhone latency).

SFCNN also has good scalability and transferability. It outperforms existing state-of-the-art lightweight models in ImageNet-1K image classification. Under 1.0G-2.0G FLOPs, SFCNN obtains +0.1% accuracy compared to SwiftFormer (Shaker et al., 2023) with 87% FLOPs. For smaller scale, SFCNN is better than UniRepLKNet (Ding et al., 2024) (79.1% vs 78.6%) with fewer FLOPs (0.7G vs 0.9G). In addition, it outperforms state-of-the-art CNNs and ViTs on dense prediction tasks, including MS-COCO instance segmentation and ADE20K semantic segmentation. In particular, SFCNN outperforms previous state-of-the-art models by a large margin (around 0.8% $Ap^b$ or 0.6% mIoU). The experimental results of our simple architecture demonstrate its great potential in vision tasks.

| Type | Reference | Method | Kernel | Param | FLOPs | Top-1 (%) |
|------|-----------|--------|--------|-------|-------|-----------|
|      | ICML21 | NFNet | | 72M | 12.4G | 83.6 |
| SK | ICLR23 | RepOpt-VGG | | 118M | 32.8G | 83.1 |
|      | CVPR21 | RegNetZ | 3 | 95M | 15.9G | 84.0 |
|      | CVPR24 | DeepMAD | | 89M | 15.4G | 84.0 |
|      | ICLR23 | RevCol | | 138M | 16.6G | 84.1 |
|      | ICLR22 | DWNet | 7 | 74M | 12.9G | 83.2 |
|      | CVPR22 | ConvNeXt | 7 | 89M | 15.4G | 83.8 |
| LK | NeurIPS22 | HorNet | 7 | 50M | 8.7G | 84.0 |
|      | ICLR23 | ConvNeXt-dcls | 17 | 89M | 16.5G | 84.1 |
|      | CVM22 | VAN | 21 | 60M | 12.2G | 84.2 |
|      | TPAMI24 | ConvFormer | 7 | 57M | 12.8G | 84.5 |
|      | CVPR22 | RepLKNet | 5,31 | 79M | 15.3G | 83.5 |
|      | ICLR24 | ConvNext-1D++ | 7,31 | 90M | 15.8G | 83.8 |
|      | NeurIPS22 | FocalNet | 3,5,7 | 89M | 15.4G | 83.9 |
| MK | CVPR24 | UniRepLKNet | 3,5,7 | 56M | 9.1G | 83.9 |
|      | ICLR23 | SLaK | 5,51 | 95M | 17.1G | 84.0 |
|      | CVPR24 | InceptionNeXt | 3,11 | 87M | 14.9G | 84.0 |
|      | CVPR24 | PeLK | 13,47,49,51,101 | 89M | 18.3G | 84.2 |
|      | ICLR24 | MogaNet | 3,5,7 | 44M | 9.9G | 84.3 |
|      | ICCV23 | ParCNetV2 | 7,13,27,55,111 | 56M | 12.6G | 84.6 |
| SK | Our | SFCNN | **3** | 49M | **8.7G** | **84.6** |

Table 1: Comparison of various CNN-based architectures on ImageNet-1K image classification. **SK** is the abbreviation of Small Kernel. **LK** is the abbreviation of Large Kernel. **MK** is the abbreviation of Multi Kernel. The top two types use the same kernel size convolution in all blocks. The second type uses several kernel sizes to process objects with variable input scales, leading to complex settings for these hyper-parameters. SK requires huge computation complexity to achieve high performance. LK and MK introduce large kernel convolution to obtain better results with fewer FLOPs, but the minimum kernel size is 7 and the largest is 111. Our SFCNN obtains the best result with the least FLOPs and only $3 \times 3$ kernel size.

Our contributions can be summarized below:

- We introduce a small kernel CNN architecture named Simple and Fast CNN, which employs a thin and deep architecture to capture more spatial information. A novel bottleneck with two $3 \times 3$ DWConvs is also proposed to enlarge the receptive field further.

- A Global Sigmoid Linear Unit activation function is proposed to capture global visual cues, which leads to richer spatial feature extraction.

- Extensive experiments demonstrate that SFCNN outperforms the state-of-the-art CNNs and ViTs in various vision tasks, including image classification, lightweight image classification, instance segmentation, and semantic segmentation.

## 2 RELATED WORK

**Convolutional Neural Network Architectures.** The introduction of AlexNet (Krizhevsky et al., 2012) marked a significant milestone in the rapid development of Convolutional Neural Networks (CNNs), with subsequent architectures (Szegedy et al., 2015; He et al., 2016; Szegedy et al., 2017) continually pushing the boundaries of performance. One recent trend in CNNs is the utilization of large convolutional kernels to achieve larger receptive fields and capture more long-range information. ConvNeXt (Liu et al., 2022) has made a noteworthy discovery, revealing that scaling the kernel size from $3\times3$ to $7\times7$ significantly contributes to performance. Similarly, DWNet (Han et al., 2022) has reached a similar conclusion by replacing the local attention layer in Swin (Liu et al., 2021) with a $7\times7$ DWConv. Following this large kernel design, some novel methods, such as VAN (Guo et al., 2023), RepLKNet (Ding et al., 2022), ConvNeXt-1d++ Kirchmeyer & Deng (2023), SLaK (Liu et al., 2023), PeLK (Chen et al., 2024), and ParCNetV2 (Xu et al., 2023), have also demonstrated impressive outcomes in many vision tasks, employing even larger kernel sizes from 21 to even 111.

Other architectures, like InceptionNeXt (Yu et al., 2024), FocalNet (Yang et al., 2022), and UniRep-pLKNet (Ding et al., 2024), and MogaNet (Li et al., 2024) combine large kernel and small kernel is one block to introduce multi-scale information, However, these methods introduce complicated architecture to employ large kernels. In addition, using large kernels or multi-branch structures will increase training difficulty and is unfriendly to hardware, resulting in longer training and inference times. Our SFCNN is a simple and fast architecture with pure $3 \times 3$ DWConv, thus obtaining an ideal speed and accuracy tradeoff.

**Transformer-based Architectures.** Transformers (Vaswani et al., 2017) have made significant breakthroughs in computer vision tasks. ViT (Dosovitskiy et al., 2021) first introduces a pure Transformer architecture for visual representations. However, directly applying self-attention to vision tasks leads to large computational costs, which is unacceptable for dense prediction tasks. Swin (Liu et al., 2021) solves this problem by utilizing window-based multi-head self-attention (MHSA) for effective feature extraction. PVT (Wang et al., 2021) proposes the pyramid hierarchical structure to extract spatial features at lower resolution. SMT (Lin et al., 2023) introduces multi-scale DWConv in one block, to avoid detail missing and retain more spatial information by information fusion across different heads in MHSA. BiFormer (Zhu et al., 2023) uses dynamic sparse attention via bi-level routing to allocate computations more flexibly. However, compared to CNNs, ViTs face hardware compatibility limitations that restrict their wider application (Zhang et al., 2023). Our SFCNN has a large receptive field with only small kernel convolutions, thus obtaining better accuracy, fewer computations, and faster speed.

# 3 METHOD

## 3.1 OVERALL ARCHITECTURE

The overall architecture of our proposed SFCNN is shown in Figure 2. Assume the size of the input image is $H \times W \times 3$, we first leverage $3 \times 3$ convolution layer with stride 2 to obtain $\frac{H}{2} \times \frac{W}{2}$ feature maps, and the dimension of the feature maps is $C$ (In SFCNN-Tiny, $C = 24$). We build a hierarchical representation with four stages. In the $i^{th}$ stage, we stack $N_i$ SFCNN blocks (In SFCNN-Tiny, $N_1 = 4, N_2 = 8, N_3 = 20, N_4 = 4$). We apply downsampling operations in the block at the beginning of each stage to reduce the resolution of the feature maps to half of the original one. Therefore, the output feature maps of the $i^{th}$ stage is $\frac{H}{2^{i+1}} \times \frac{W}{2^{i+1}}$. We stack more $3 \times 3$ convolutions in one SFCNN block and design a thinner and deeper architecture compared with ConvNeXt (Liu et al., 2022), to enlarge the receptive field. We also propose a Global Sigmoid Linear Unit (GSiLU) activation function to capture global spatial information.

## 3.2 COMPUTING THE RECEPTIVE FIELD

The ultimate objective of introducing large kernel convolution is to increase the receptive field. For a convolution with $L$ layers, feature map $f_l \in \mathbf{R}^{c_l \times h_l \times w_l}, l = 1, 2, ..., L$ denotes the output of the $l$-th layer, with channel $c_l$, height $h_l$, and width $w_l$. We denote the input image by $f_0$, and the final output feature map corresponds to $f_L$. Each layer $l$'s spatial configuration is parameterized by kernel size $k_l$ and stride $s_l$. Define $r_l$ as the receptive field size of $l$-th layer, we give a simplified equation from Araujo et al. (2019) to compute the receptive field:

$$r_l = r_{l-1} + (k_l - 1) \cdot \sum_{i=1}^{l-1} s_l. \tag{1}$$

According to this equation, increasing the kernel size and stride is feasible to enlarge the receptive field. However, we have also noticed it is a recurrence equation, increasing the number of recursion iterations could also increase the receptive field, which means adding more DWConvs.

## 3.3 SIMPLE AND FAST CONVOLUTIONAL NEURAL NETWORK BLOCK

In this section, we design the SFCNN block, which uses more $3 \times 3$ DWConvs. As shown in Figure 2, we design two types of SFCNN blocks. One is a common block, and another is equipped with an additional downsampling operation. We design the SFCNN block as follows step by step:

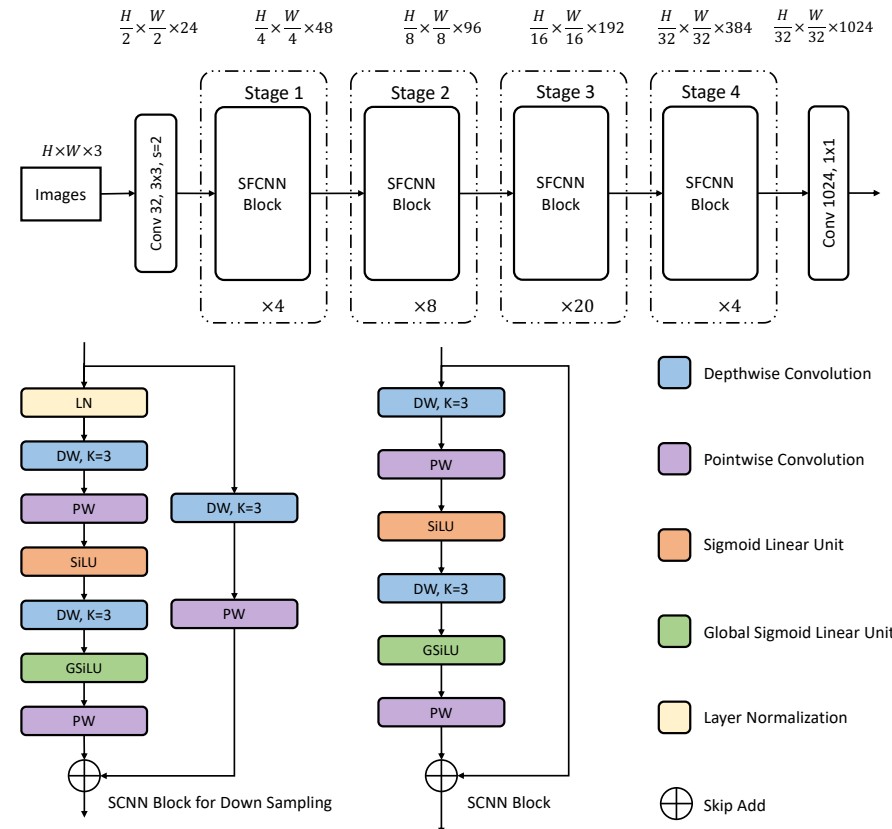

Figure 2: **The architecture of SFCNN-Tiny.** It mainly consists of our well-designed SFCNN block. In addition, we design a variant for downsampling instead of introducing a convolution with stride 2 as patch merging in ConvNeXt (Liu et al., 2022).

1. We apply a $3 \times 3$ DWConv for input features to capture spatial information.

2. The output feature of step 1 passes through a pointwise convolution (PWConv) and a Sigmoid Linear Unit (SiLU) to exchange channel information and obtain nonlinearity.

3. The output feature of step 2 is sent to another $3 \times 3$ DWConv to capture more visual cues.

4. The output feature of step 3 passes through a Global Sigmoid Linear Unit (GSiLU) to capture global coarse-grained information.

5. The output feature of step 4 is sent to a PWConv to exchange channel information again.

6. As for the common block, the input of step 1 and the output features of step 5 are added together to enhance network expressiveness and alleviate the gradient vanishing.

7. As for the downsampling block, the input of step 1 will go through a $3 \times 3$ DWConv with stride 2, a PWConv, and then be added with the features of step 5.

The SFCNN block achieves a large receptive field by stacked $3 \times 3$ DWConvs and avoids the issues brought by large kernel sizes, such as the extra time in training and deployment. The receptive field of two $3 \times 3$ DWConvs is the same as one $5 \times 5$ convolution (Zhang et al., 2023), so our design can reduce the difficulty of training and deployment brought about by the use of many large convolution kernels, and remain large receptive field information.

### 3.4 Thin and Deep Architecture

Inceptionv3 (Szegedy et al., 2016) points out that a multilayer network could replace a large kernel convolution with less computation complexity, and its experimental results prove this. Equation 1

| model | FLOPs | Input Resolution | Stage 1 | | Stage 2 | |
|-------|-------|------------------|---------|-----------------|---------|-----------------|
|       |       |                  | Number | Receptive Field | Number | Receptive Field |
| W640  | 2.47G |                  | 1 | $21 \times 21$ | 3 | $117 \times 117$ |
| W576  | 2.44G |                  | 2 | $37 \times 37$ | 4 | $165 \times 165$ |
| W512  | 2.49G | $224 \times 224$ | 2 | $37 \times 37$ | 5 | $197 \times 197$ |
| W384  | 2.44G |                  | 4 | $69 \times 69$ | 8 | $325 \times 325$ |

Table 2: SFCNN-Tiny is the baseline model, the same as W384, which means the dimensions are set to 48, 96, 192, and 384 respectively. We reduce the block number of all stages proportionally to design three heavy and shallow models with similar FLOPs.

also shows that more spatial convolution is one of the key factors in the receptive field. Motivated by these, we design a thin and deep model with more $3 \times 3$ DWConv instead of a heavy and shallow model with a large kernel convolution. As shown in Table 2, we design four tiny models with different depths and widths. In the ImageNet dataset (Deng et al., 2009), the input resolution is often set to $224 \times 224$. The receptive field of the deepest model W384 is even almost triple the size of the shallowest W640. In particular, the receptive fields of W384 in stage two are larger than the input resolution, which means that it has a global receptive field, while other shallow models only have a local one.

### 3.5 GLOBAL SIGMOID LINEAR UNIT

Sigmoid Linear Unit (SiLU) is a widely used activation function, which was originally coined in GELU (Hendrycks & Gimpel, 2016), and later works (Ramachandran et al., 2018; Elfwing et al., 2018) demonstrate its effectiveness. After GPT using GELU, many subsequent models follow it by default, including recent ViTs (Liu et al., 2021) and MLPs (Lai et al., 2023). GELU can be approximated as

$$GELU(x) = x \times \Phi(x) \approx 0.5 \times x \times (1 + \tanh(\sqrt{2/\pi}) \times (x + 0.044715 \times x^3)), \quad (2)$$

where $\Phi$ means the cumulative distribution function for the Gaussian distribution. Another approximate formula for GELU is:

$$GELU(x) \approx x \times \sigma(1.702 \times x), \quad (3)$$

where $\sigma$ is a sigmoid function. Similarly, Swish (Ramachandran et al., 2018) proposes to take advantage of automatic search techniques to discover a new activation function named Swish, which can be formulated as

$$Swish(x) = x \times \sigma(\beta \times x). \quad (4)$$

It is easy to see that Swish has a similar formulation of GELU. The difference is that the learnable parameter in Swish is set to a fixed value of 1.702. Meanwhile, in reinforcement learning, to achieve the same goal of output from one hidden unit in the expected energy restricted Boltzmann machine (EE-RBM), SiLU (Elfwing et al., 2018) proposes an activation function for the approximation of neural network functions:

$$SiLU(x) = x \times \sigma(x). \quad (5)$$

SiLU is a simplified version of Swish and GELU, and it does not require a learnable parameter or a fixed value inside the sigmoid function. However, SiLU computes the results in all positions individually. It is unable to capture spatial information. We hope it achieves a global receptive field to let our SFCNN closer to those large-kernel CNNs. Thus, we propose a Global Sigmoid Linear Unit (GSiLU) activation function to capture global spatial visual cues. The formula is as follows:

$$GSiLU(x) = x \times \sigma(GAP(x)), \quad (6)$$

where GAP is a global average pooling operation. It embeds global information from every channel into a single value to produce the importance of these channels.

However, GSiLU is very similar to the famous Squeeze-and-Excitation (Hu et al., 2018) module, but considering the huge extra parameter as shown in Table 8, we use GSiLU because it is a non-parametric module.

| Family | Reference | Method | Param | FLOPs | Top-1 (%) |
|---|---|---|---|---|---|
| ViT | ICCV21 | Swin-T Liu et al. (2021) | 29M | 4.5G | 81.3 |
| | | Swin-S Liu et al. (2021) | 50M | 8.7G | 83.0 |
| | | Swin-B Liu et al. (2021) | 88M | 15.4G | 83.5 |
| | CVPR23 | BiFormer-T Zhu et al. (2023) | 13M | 2.2G | 81.4 |
| | | BiFormer-S Zhu et al. (2023) | 26M | 4.5G | 83.7 |
| | | BiFormer-B Zhu et al. (2023) | 58M | 9.8G | 84.3 |
| | ICCV23 | SMT-T Lin et al. (2023) | 12M | 2.4G | 82.2 |
| | | SMT-S Lin et al. (2023) | 21M | 4.7G | 83.7 |
| | | SMT-B Lin et al. (2023) | 32M | 7.7G | 84.3 |
| CNN | ICLR22 | DWNet Han et al. (2022) | 24M | 3.8G | 81.3 |
| | | DWNet Han et al. (2022) | 74M | 12.9G | 83.2 |
| | CVPR22 | ConvNeXt-T Liu et al. (2022) | 29M | 4.5G | 82.1 |
| | | ConvNeXt-S Liu et al. (2022) | 50M | 8.7G | 83.1 |
| | | ConvNeXt-B Liu et al. (2022) | 89M | 15.4G | 83.8 |
| | ICLR23 | SLaK-T Liu et al. (2023) | 30M | 5.0G | 82.5 |
| | | SLaK-S Liu et al. (2023) | 55M | 9.8G | 83.8 |
| | | SLaK-B Liu et al. (2023) | 95M | 17.1G | 84.0 |
| | ICCV23 | ParCNetV2-T Xu et al. (2023) | 25M | 4.3G | 83.5 |
| | | ParCNetV2-S Xu et al. (2023) | 39M | 7.8G | 84.3 |
| | | ParCNetV2-B Xu et al. (2023) | 56M | 12.5G | 84.6 |
| | CVPR24 | PeLK-T Chen et al. (2024) | 29M | 5.6G | 82.6 |
| | | PeLK-S Chen et al. (2024) | 50M | 10.7G | 83.9 |
| | | PeLK-B Chen et al. (2024) | 89M | 18.3G | 84.2 |
| | ICLR24 | MogaNet-S Li et al. (2024) | 25M | 5.0G | 83.4 |
| | | MogaNet-B Li et al. (2024) | 44M | 9.9G | 84.3 |
| | | MogaNet-L (Li et al., 2024) | 83M | 15.9G | 84.7 |
| | CVPR24 | UniRepLKNet-N Ding et al. (2024) | 18M | 2.8G | 81.6 |
| | | UniRepLKNet-T Ding et al. (2024) | 31M | 4.9G | 83.2 |
| | | UniRepLKNet-S Ding et al. (2024) | 56M | 9.1G | 83.9 |
| | CVPR24 | InceptionNeXt-T Yu et al. (2024) | 28M | 4.2G | 82.3 |
| | | InceptionNeXt-S Yu et al. (2024) | 49M | 8.4G | 83.5 |
| | | InceptionNeXt-B Yu et al. (2024) | 87M | 14.9G | 84.0 |
| | **Our** | SFCNN-T | 16M | 2.4G | 82.6 |
| | | SFCNN-S | 27M | 4.5G | 83.7 |
| | | SFCNN-B | 49M | 8.7G | 84.6 |
| | | SFCNN-B$^{256\times256}$ | 49M | 11.4G | 84.8 |
| | | SFCNN-B$^{288\times288}$ | 49M | 14.5G | 84.9 |

Table 3: Comparison with other SOTA models on ImageNet-1K classification.

## 3.6 ARCHITECTURE VARIANTS

We set different numbers of blocks in Stage $1 \sim 4$ as $\{S_1, S_2, S_3, S_4\}$, and expand the channel dimensions as shown in Figure 2 to obtain variants of the SFCNN architecture. By balancing performance and inference time, we design five versions of our models as follows:

- SFCNN-P (Pico): $C$=32, block numbers={3,4,12,3}, expand ratio=4

- SFCNN-N (Nano): $C$=40, block numbers={3,6,17,3}, expand ratio=4

- SFCNN-T (Tiny): $C$=48, block numbers={4,8,20,4}, expand ratio=4

- SFCNN-S (Small): $C$=64, block numbers={6,12,28,6}, expand ratio=3

- SFCNN-B (Base): $C$=80, block numbers={8,15,35,8}, expand ratio=3

The parameters (model size), FLOPs (computation complexity), and top-1 accuracy on ImageNet-1K of the variants of SFCNN architecture are shown in Table 3.

| Family | Reference | Method | Param | FLOPs | Top-1 (%) |
|--------|-----------|--------|-------|-------|-----------|
| ViT | ICCV23 | FastViT-T8 | 3.6M | 0.7G | 75.6 |
| ViT | NeurIPS23 | FAT-B0 | 4.5M | 0.7G | 77.6 |
| ViT | ICCV23 | SwiftFormer-S | 6.1M | 1.0G | 78.5 |
| CNN | ICLR2024 | MogaNet-XT | 3.0M | 1.0G | 77.2 |
| CNN | CVPR2024 | UniRepLKNet-F | 6.2M | 0.9G | 78.6 |
| CNN | **Our** | SFCNN-P | 7.7M | 0.7G | **79.1** |
| ViT | ICCV2023 | FastViT-SA12 | 10.9M | 1.9G | 80.6 |
| ViT | NeurIPS23 | FAT-B1 | 7.8M | 1.2G | 80.1 |
| ViT | ICCV23 | SwiftFormer-L3 | 12.1M | 1.6G | 80.9 |
| CNN | ICLR2024 | MogaNet-T | 5.2M | 1.4G | 80.0 |
| CNN | CVPR2024 | UniRepLKNet-P | 10.7M | 1.6G | 80.2 |
| CNN | **Our** | SFCNN-N | 11.1M | 1.4G | **81.0** |

Table 4: Comparison with other lightweight models on ImageNet-1K. SFCNN-P and SFCNN-N are compared with other lightweight models with less than and more than 1G FLOPS, respectively.

## 4 EXPERIMENTS

In this section, starting with the evaluation of SFCNN in the ImageNet-1K dataset Deng et al. (2009) for image classification, we subsequently expand our assessment of MS-COCO Lin et al. (2014) instance segmentation, as well as ADE20K Zhou et al. (2017) semantic segmentation.

### 4.1 IMAGENET-1K CLASSIFICATION

**Experimental Setup.** To evaluate the effectiveness of our SFCNN, we utilize the ImageNet-1K Deng et al. (2009) dataset, which consists of 1.2 million training images and 50,000 validation images across 1,000 categories. Our primary metric for experimentation is the top-1 accuracy. During the training phase, we use the AdamW optimizer with a batch size of $1024$ and initialize the learning rate at $0.001$. To facilitate learning, we incorporate cosine decay and introduce a weight decay of 0.05. The training process spans 300 epochs, with a warm-up strategy implemented for the initial 20 epochs. For data augmentation and regularization, we adopt the same strategies as ConvNeXt Liu et al. (2022).

**Comparison with SOTA Models.** Table 3 compares SFCNNs with state-of-the-art CNNs and ViTs. Our methods demonstrate superior performance compared to SMT Lin et al. (2023), MogaNet Li et al. (2024), and UniRepLKNet Ding et al. (2024). In particular, our SFCNN-N achieves a higher top-1 accuracy of 82.6% (compared to 82.2%) compared to SMT-T with the same FLOPs (4.5G). Additionally, our small version of SFCNN achieves better results than Swin Transformer Liu et al. (2021) while requiring only approximately 30% computation. Compared with MogaNe-B, our base version achieves better accuracy (84.6% vs 84.3%) with fewer FLOPs (8.7G vs. 9.9G).

**Comparison with lightweight Models.** Table 4 compares SFCNNs with state-of-the-art lightweight CNNs and ViTs. Our pico version is better than the sota CNN UniRepLKNet (+0.5%) with fewer FLOPs (-0.2G). The nano version also obtains better results (+0.1% top-1) with 88% FLOPs compared to SOTA ViT SwiftFormer-L3.

### 4.2 INSTANCE SEGMENTATION ON COCO

**Experimental Setup.** We conduct instance segmentation employing Mask-RCNN as the framework. MS-COCO Lin et al. (2014) dataset is selected, with 118k training data and 5k validation data. We compare SFCNN with other backbones. All Hyperparameters align with Swin Transformer: AdamW optimizer, learning rate of 0.0003, weight decay of 0.05, and batch size of 4 images/GPU (8 GPUs). We use a multi-scale training strategy. Backbones are initialized with ImageNet-1K pre-trained weights. Models are trained for 36 epochs with a $3\times$ schedule.

**Results.** The performance of our SFCNN on the COCO dataset is presented in Table 5, along with other architectures. Our proposed SFCNN achieves superior results to the Swin Transformer and

| Backbone | $AP^b$ | $AP^b_{50}$ | $AP^b_{75}$ | $AP^m$ | $AP^m_{50}$ | $AP^m_{75}$ | Params | FLOPs |
|---|---|---|---|---|---|---|---|---|
| Mask R-CNN (3×) | | | | | | | | |
| ResNet50 He et al. (2016) | 41.0 | 61.7 | 44.9 | 37.1 | 58.4 | 40.1 | 44M | 260G |
| PVT-S Wang et al. (2021) | 43.0 | 65.3 | 46.9 | 39.9 | 62.5 | 42.8 | 44M | 245G |
| AS-MLP-T Lian et al. (2022) | 46.0 | 67.5 | 50.7 | 41.5 | 64.6 | 44.5 | 48M | 260G |
| Hire-MLP-S Guo et al. (2022) | 46.2 | 68.2 | 50.9 | 42.0 | 65.6 | 45.3 | - | 256G |
| Swin-T Liu et al. (2021) | 46.0 | 68.2 | 50.2 | 41.6 | 65.1 | 44.9 | 48M | 267G |
| ConvNeXt-T Liu et al. (2022) | 46.2 | 67.9 | 50.8 | 41.7 | 65.0 | 44.9 | 48M | 267G |
| SFCNN-S (ours) | **47.8** | **69.2** | **52.6** | **43.0** | **66.6** | **46.2** | 42M | 252G |
| ResNet101 He et al. (2016) | 42.8 | 63.2 | 47.1 | 38.5 | 60.1 | 41.3 | 63M | 336G |
| PVT-Medium Wang et al. (2021) | 44.2 | 66.0 | 48.2 | 40.5 | 63.1 | 43.5 | 64M | 302G |
| AS-MLP-S Lian et al. (2022) | 47.8 | 68.9 | 52.5 | 42.9 | 66.4 | 46.3 | 69M | 346G |
| Hire-MLP-B Guo et al. (2022) | 48.1 | 69.6 | 52.7 | 43.1 | 66.8 | 46.7 | - | 335G |
| Swin-S Liu et al. (2021) | 48.5 | 70.2 | 53.5 | 43.3 | 67.3 | 46.6 | 69M | 359G |
| SFCNN-B (ours) | **49.3** | **70.7** | **54.4** | **44.3** | **68.0** | **48.0** | 64M | 334G |

Table 5: The instance segmentation results of different backbones on the COCO dataset.

| Method | Backbone | val MS mIoU | Params | FLOPs |
|---|---|---|---|---|
| UperNet Xiao et al. (2018) | Swin-T Liu et al. (2021) | 45.8 | 60M | 945G |
| | AS-MLP-T Lian et al. (2022) | 46.5 | 60M | 937G |
| | ConvNeXt-T Liu et al. (2022) | 46.7 | 60M | 939G |
| | Hire-MLP-S Guo et al. (2022) | 47.1 | 63M | 930G |
| | InceptionNeXt-T Yu et al. (2024) | 47.9 | 56M | 933G |
| | SFCNN-S (ours) | **48.8** | 54M | 938G |
| UperNet Xiao et al. (2018) | Swin-S Liu et al. (2021) | 49.5 | 81M | 1038G |
| | AS-MLP-S Lian et al. (2022) | 49.2 | 81M | 1024G |
| | ConvNeXt-S Liu et al. (2022) | 49.6 | 82M | 1027G |
| | Hire-MLP-B Guo et al. (2022) | 49.6 | 88M | 1011G |
| | InceptionNeXt-S Yu et al. (2024) | 50.0 | 78M | 1020G |
| | SFCNN-B (ours) | **50.6** | 75M | 1025G |

Table 6: The semantic segmentation results of different backbones on the ADE20K validation set.

| PreConv | MidConv | PreGSiLU | MidGSiLU | Top-1 (%) | Param | FLOPs | Activation | Top-1 (%) | Param | FLOPs |
|---|---|---|---|---|---|---|---|---|---|---|
| | ✓ | | | 81.6 | 16M | 2.39G | ReLU | 82.0 | 16M | 2.44G |
| ✓ | | | | 81.8 | 15M | 2.34G | PReLU | 82.1 | 16M | 2.44G |
| ✓ | ✓ | | | 82.0 | 16M | 2.43G | SiLU | 82.3 | 16M | 2.44G |
| ✓ | ✓ | ✓ | | 82.2 | 16M | 2.44G | GELU | 82.3 | 16M | 2.44G |
| ✓ | ✓ | | ✓ | **82.6** | 16M | 2.44G | GSiLU | 82.6 | 16M | 2.44G |
| ✓ | ✓ | ✓ | ✓ | 82.6 | 16M | 2.45G | SE | 82.7 | 25M | 2.46G |

Table 7: Ablation analysis on the convolution and activation. Table 8: Ablation analysis on the ac-
Pre and Mid mean the first and second units of two DWConvs tivation. GSiLU could be regarded as
in the block.                                                a variant of SE without parameters.

requires fewer FLOPs. Specifically, Mask R-CNN + Swin-S achieves an $AP^b$ of 48.5 with 359
GFLOPs, whereas Mask R-CNN + SFCNN-B achieves an $AP^b$ of 49.3 with 334 GFLOPs.

## 4.3 SEMANTIC SEGMENTATION ON ADE20K

**Experimental Setup.** We use the UperNet Xiao et al. (2018) framework to evaluate our methods
on ADE20K Zhou et al. (2017). In training, we initialize the backbone with ImageNet weights and
use Xavier initialization for other layers. AdamW optimizer with initial learning rate $1.0 \times 10^{-4}$
is used. Training involves 160k iterations, batch size 16 on 8×A100 GPUs, weight decay 0.01,
and polynomial decay schedule with power 0.9. Data augmentation includes random horizontal
flipping, rescaling (0.5-2.0), and photometric distortion. The stochastic depth ratio is set to 0.3. The
evaluation metric is multi-scale mean Intersection over Union (MS mIoU).

**Result.** Table 6 presents a performance comparison between our SFCNN and state-of-the-art ar-
chitectures on the ADE20K dataset. Despite having similar FLOPs, SFCNN-T achieves superior
results to Swin-T, with an MS mIoU of 48.4 versus 45.8.

| block numbers | channel dims | Params | FLOPs | Top-1 |
|---|---|---|---|---|
| 1,3,7,1 | 80,160,320,640 | 14M | 2.47G | 81.3 |
| 2,4,7,2 | 72,144,288,576 | 15M | 2.44G | 81.8 |
| 2,5,11,2 | 64,128,256,512 | 15M | 2.49G | 82.2 |
| 4,8,20,4 | 48,96,192,384 | 16M | 2.44G | **82.6** |
| 6,12,28,6 | 40,80,160,320 | 16M | 2.44G | 82.4 |

Table 9: Ablation analysis on the model depth with similar complexity. Block numbers mean the numbers in four stages, while channel dims mean the channel dimensions in the same four stages.

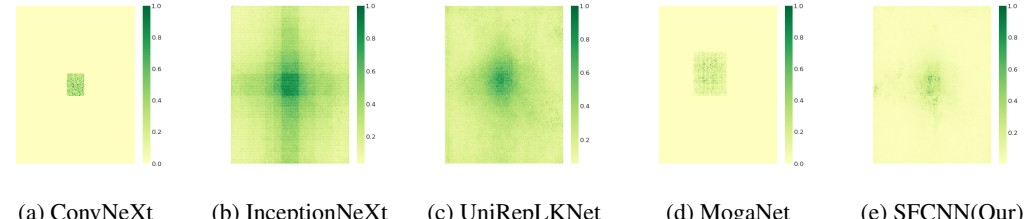

(a) ConvNeXt     (b) InceptionNeXt     (c) UniRepLKNet     (d) MogaNet     (e) SFCNN(Our)

Figure 3: **Effective receptive field (ERF) of various CNNs.** Our SFCNN could capture long-range dependency and the local context features simultaneously.

## 4.4 ABLATION STUDY

**The Impact of DWConv.** As shown in Table 7 lines one to three, the result is markedly declined when we remove one DWConv in the SFCNN block. The receptive field will become almost halved by using only one DWConv in a block.

**The Impact of GSiLU.** As shown in Table 7 lines four to six, adding GSiLU could bring at least +0.2% top-1 accuracy, but adding two GSiLU obtains the same performance as the one. One GSiLU could capture enough global spatial information for a single block.

**The Impact of Activation.** In table 8, we replace the GSiLU with other widely used activation and the SE module. GSiLU could obtain better results with the same FLOPs and parameters. This proves the importance of capturing long-range visual cues because SiLU only uses original feature maps as input, while GSiLU could capture global spatial information. SE is better than GSiLU, but it introduces extra huge parameters, thus we choose GSiLU because it is a non-parametric module.

**The Impact of Model Depth.** Table 9 shows five models with different depths. A thinner and deeper architecture could obtain better results than heavier and shallower models. The main reasons may be a larger receptive field and better non-linear fitting capability. However, the deepest model has a much thinner channel width, which will lose information and even get a -0.2% performance.

**Visualization of the Receptive Field.** Figure 3 visualizes the receptive field of many CNNs. Our SFCNN could capture long-range dependency and the local context features simultaneously, while other CNNs only capture local information or introduce global noises. ParCNetV2 only provides the code but does not provide the pre-train weight, thus we cannot visualize it.

## 5 CONCLUSION

We propose the Simple and Fast Convolutional Neural Network (SFCNN) that mainly employs a sequence of stacked $3 \times 3$ convolutions to capture visual cues of various sizes. Though the architecture is simple, SFCNN surpasses the state-of-the-art CNNs with larger kernels. SFCNN is a thin and deep model, encouraging more layers of DWConv to capture more spatial information under the same computing complexity. Furthermore, we redesign the traditional inverted residual bottleneck with two DWConv to enlarge the receptive field. We propose a novel Global Sigmoid Linear Unit (GSiLU) activation function to capture global coarse-grained spatial information. SFCNN achieves the best accuracy in ImageNet-1K image classification based on four evaluations of computational complexity. Besides, experimental results on lightweight image classification, instance segmentation, and semantic segmentation further verify the superiority of SFCNN.

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

## A APPENDIX

You may include other additional sections here.

