# OpenReview forum: "Simple and Fast CNN for Vision"
_ICLR.cc/2025/Conference — Submitted to ICLR 2025_

### Official Review · Reviewer_fTbR · 2024-10-26

**Soundness:** 3
**Presentation:** 4
**Contribution:** 1
**Rating:** 3
**Confidence:** 5

**Summary:**

This paper is a very traditional architecture design paper. The major motivation of this work is: large kernel vs small kernel. This debate has been in the community since 10 AlexNet. In VGG, they change the large kernel from 7x7 to a stack of 3x3. Recent years observe a reverse trend that moving back to extremely large kernels. In this work, the authors argue again to use stack of small kernels, due to "Nevertheless, these approaches are unfriendly to hardware, imposing a serious computation burden on training or inference"

Based on this motivation, the authors carefully craft a new architecture, SFCNN. The architecture shows minor performance gain on IN1k ("+0.1% accuracy compared to SwiftFormer (Shaker et al., 2023) with 87% FLOPs"), as well as downstream tasks like COCO and ADE20k.

**Strengths:**

1. The paper writing is clear, including the motivation, main results summary, architecture design, main results, architecture ablations. All these necessary components are easy to find and comprehend.

2. The experiments are adequate for a traditional architecture design work, including main results on 1N1k, coco and ADE20k. There are also component contribution ablations and architecture variant.

**Weaknesses:**

1. The motivation / idea of this work is not new (from large kernels to a stack of smaller kernels.)

This idea dates back to VGG (2014). Authors can refer to sec 2.3 in the paper for more discussions. Placing this as the main motivation largely harm the overall contribution, because this makes the paper more like a revisit / conversation in the debate.

2. Minor performance gain vs large variance in different architecture hyperparams.

The performance gain over SOTA models is minor, compared with performance variance in similar architectures with different hyperparams. As shown in Table9, searching a best setup for 1N1k is critical (min 81.3 vs max 82.6), while the performance gain over sota is only 0.x% level. This is also reflected in Table 8.

I deeply appreciate the efforts in searching a best setup for the architecture. However, this makes the major performance contribution more in the "searching" part but no in the architecture itself. Currently, due to the development of NAS, such searching efforts can be largely automated.

3. (Minor) Table 7 and 8 are mixed together in the manuscript. It is confused.

**Questions:**

1. Minor performance gain vs large variance in different architecture hyperparams. This deserves a deep discussion.

2. Due to the nature of carefully manually crafted CNN (which may be overfitted on IN1k), I am wondering how the architecture perform on IN22k-pretraining + IN1k-finetuning?

*This is not a must-do due to the training cost. However, if this is provided, my concern on the performance perspective can be alleviated.

**Details Of Ethics Concerns:**

N.A.

---

### Official Review · Reviewer_d9Q7 · 2024-10-28

**Soundness:** 3
**Presentation:** 3
**Contribution:** 2
**Rating:** 5
**Confidence:** 5

**Summary:**

This paper presents a new convolutional neural network architecture, called SFCNN. Unlike recent popular CNN works that mainly aim to explore how to better take advantage of large-kernel convolutions, this paper explains that using thin but deep network architecture with only 3x3 convolutions can still achieve good results. In addition, the authors also rethink of the design of the SiLU activation function and propose a new one, which involves in global information based on SiLU. Experiments show that the classification performance on ImageNet is better than most previous CNN-based models. In terms of latency, the proposed approach achieves better results as well.

**Strengths:**

- The motivation of this paper is clear. Using small-kernel convolutions may lead to faster inference but makes the receptive field of CNNs not large enough to capture the target objects. This paper presents to add more 3x3 convolutions to enlarge the receptive field of the proposed network.

- This paper receives better trade-off between model classification performance and latency. Compared to recent CNN-based models, the proposed method has compact model architecture but better performance.

**Weaknesses:**

- The authors claim that large receptive field is important for CNNs. In Fig. 3, it is shown that the proposed approach has large effective receptive field. However, it is not as large as the one of UniRepLKNet. According to the numerical results on ImageNet, the proposed approach gets better numbers. Does this mean that large effective RF is not an important measurement for building CNNs?

- The authors claim that their bottleneck block design with two 3 × 3 DWConvs is novel. However, as far as I know, adding two depthwise convolutions in a single basic block has been explored before, e.g., MobileNeXt (ECCV'20, Zhou et al.). Though the order of the layers is a bit different, the design intention is similar. So, I do not think this can be viewed as a core contribution for this paper.

- From the paper, it seems that CNNs with thin but deep architecture and small kernel convolutions perform more efficient than those with large kernels. However, the macro model architecture of the proposed method is not actually the same to previous large-kernel CNNs. I think the authors should conduct more specific experiments to demonstrate this.

- In Table 5, it is good to see the results on instance segmentation but the methods the authors compare with are not new already. I have no idea why the results of the recent published works on CNNs are not reported here.

- It seems that the 7th table has two captions? Where is Table 8?

- From the ablation results,  it seems that the proposed GSiLU indeed performs better than other activation functions. However, have the authors analyzed why global information should be added into activation functions? The motivation of designing such an activation function is not clear. In addition, as GSiLU is already used, why the original SiLU is still used?

**Questions:**

- The contributions of this paper should be further explained.

- More analysis on the advantages of using multiple small-kernel convolutions should be elaborated more.

- The motivation of introducing global information in activation functions should be made clearer.

---

### Official Review · Reviewer_pTXi · 2024-11-02

**Soundness:** 2
**Presentation:** 2
**Contribution:** 2
**Rating:** 5
**Confidence:** 5

**Summary:**

This paper proposes a CNN architecture for visual recognition. The main contribution is a small CNN architecture called Simple and Fast CNN with a core idea of stacking 3x3 convolutions to design a deep architecture. The work proposes an inverted residual bottleneck with two 3x3 depth-wise convolutions. Also, this paper proposes a Global Sigmoid Linear Unit activation function to capture global information.

**Strengths:**

1. The paper performed experiments of the proposed approach on several visual recognition tasks.
2. The architecture presents good results in comparison with other approaches.

**Weaknesses:**

1. The main concern is regarding the lack of novelty. The proposed contributions are already well known and explored in literature for quite a long time. For instance, using a sequence of stacked 3x3 convolutions to enlarge the receptive field is an approach deeply explored in computer vision community for many years (ResNets, VGG-Nets, MobileNets, etc). Depth-wise convolutions are also explored intensively for efficiency gains (Xception, MobileNet, etc). Including the proposed Global Sigmoid Linear Unit is just a form of the existing work (already quite old) Squeeze-and-Excitation Networks. After reading this work I could not find anything novel or some new insight that is not already known to the vision community.
2. Besides lack of novelty, this work does not compensate with some new experimental findings or some new insights for practitioners.

Overall, I find the contributions of this work to be quite limited to qualify for publishing the work at such high venue. Maybe a workshop contribution can be more appropriate.

**Questions:**

Please see above my main concerns.

---

### Official Review · Reviewer_nywT · 2024-11-03

**Soundness:** 2
**Presentation:** 2
**Contribution:** 3
**Rating:** 5
**Confidence:** 5

**Summary:**

This paper targets the computational inefficiency and hardware compatibility issues of recent ConvNets that rely on large kernels to capture long-range dependencies. The authors propose a new ConvNet architecture SFCNN for vision tasks, which has shown impressive performance through a thin-and-deep design philosophy. Concretely, it combines a dual 3×3 depth-wise convolutions branch with Global Sigmoid Linear Unit (GSiLU) activation, which captures both local and global dependencies without large kernels. The proposed SFCNN is evaluated on mainstream vision benchmarks, such as ImageNet-1K classification, COCO instance segmentation, and ADE20K semantic segmentation, demonstrating great performance while maintaining better hardware efficiency across different platforms (GPU, TensorRT, iPhone). The experiments seem to strongly support the claims about achieving better accuracy-efficiency trade-offs compared to existing ConvNets.

**Strengths:**

**(S1) A critical research question with real-world significance:**
The paper shows great industrial relevance by addressing a critical need in real-world deployment scenarios, in which computational efficiency is crucial, particularly in edge devices and mobile applications. The proposed SFCNN is notably cost-effective, providing a more resource-efficient alternative to existing approaches while maintaining or improving performance metrics. The presented thin-and-deep architecture appears to show great scalability, demonstrating computational efficiency across different model sizes, and making it highly adaptable to various resource constraints. From a practical impact perspective, this work has the potential to significantly reduce infrastructure costs for computer vision applications at scale, making it valuable for industrial applications rather than merely pushing the limit of accuracy metrics.

**(S2) Thorough experiments and validation:**
Extensive experiments are conducted on multiple mainstream computer vision tasks, such as ImageNet-1K classification, COCO detection, and ADE20K semantic segmentation. The consistency of performance across different scales is noteworthy. Ablation studies are also conducted, providing a detailed analysis of the contribution of each component to the overall performance. More importantly, the authors present a clear demonstration of the impact of model depth vs. width, supported by the evaluation of different activation functions and receptive field analysis. Hardware performance evaluation is particularly thorough, encompassing cross-platform testing on GPU, TensorRT, and iPhone, with detailed latency and throughput measurements under various scenarios. All these experiments strongly support the paper’s claim.

**Weaknesses:**

**(W1) Technical Originality:**
The basic building blocks of SFCNN, including 3×3 depth-wise convolutions and point-wise convolutions, largely rely on well-established techniques without significant technical originality. The GSiLU activation also bears considerable similarity to existing approaches like CBAM and SE modules. The thin-and-deep philosophy, while effectively implemented, has been explored in previous works. The theoretical foundation could be strengthened significantly, as it currently lacks enough theoretical insights into the nature of convolution operations and their relationships with model depth. A more thorough analysis of the relationship between depth and receptive field would strengthen the paper's contributions.

**(W2) Technical Soundness & Empirical Analysis:**
While mobile testing is included, more empirical analysis could benefit this work and improve its technical soundness. For example, the Grad-CAM heat map visualization and training dynamics investigation would provide insightful and straightforward support for understanding the technical strengths of SFCNN. Moreover, the discussion of failure cases and limitations is inadequate, potentially leaving practitioners without clear guidance on the architecture's boundaries. The exploration of model behavior under extreme resource constraints could provide valuable insights for edge deployment scenarios. I strongly recommend that the authors carry out more empirical analyses that lead to more systematic conclusions for efficient ConvNet design. The thin-and-deep design philosophy is inspiring but not specific and systematic enough. Also, this work first tries stacking multiple depth-wise convolutions in a single block rather than just one. How it works for better representation capacity is still worth digging deep.

**(W3) Presentation Clarity and Details:**
The writing organization exhibits several points that require further improvement. The technical content sometimes lacks coherence, with important methodological details scattered across different sections rather than presented in a unified manner. The description of the architecture could benefit from a more structured approach, particularly in explaining the interaction between different components. Several key concepts are offered within dense paragraphs, making it challenging for readers to extract crucial implementation details. In addition, the method description, while comprehensive, could be reorganized to better highlight the progressive development of ideas and design choices. Moreover, the tables of experimental results presentation would benefit from highlighting the performance advantages. The formatting consistency across tables and figures needs attention, with some inconsistencies in style and presentation detracting from the overall appearance. For example, the thickness of table lines is inconsistent. I recommend the authors to first go through the entire manuscript for a thorough refinement.

**Questions:**

**(Q1) Trade-offs Analysis and Discussions:**
The paper's analysis of various trade-offs deserves deeper exploration. The proposed SFCNN shows great superiority in speed. However, I have noticed that some architectures like MogaNet show better parameter efficiency while at lower speeds. Thus, a more detailed investigation of parameter efficiency versus computational speed would provide valuable insights for practitioners choosing between different model configurations. Moreover, there are several points that are tightly associated with this work that deserve further exploration: First, the memory-compute trade-off analysis could be expanded to include different hardware scenarios and deployment conditions. Second, the relationship between training efficiency and inference efficiency deserves more attention, since these can often have different optimal choices. Third, the model scaling properties, particularly regarding the relationship between model depth and width at different computational budgets.

**(Q2) Broader Architecture Considerations:**
The scope of this paper lies in ConvNets in vision tasks. However, there are more kinds of architecture emerged these years. A thorough comparison with emerging architectures like Vision Mamba and RWKV models would provide valuable context for the field's evolution. Besides, there are various efficient computation techniques proposed to boost the computational efficiency of these new architectures. The evaluation against attention-based alternatives could provide insights into the relative strengths and weaknesses of different vision backbone architectures. These expanded analyses and discussions would significantly strengthen the soundness and contribution of this paper and provide valuable guidance for future research in the community.

---
**Additional Comment:**

I hope my review helps to further strengthen this paper and helps the authors, fellow reviewers, and Area Chairs understand the basis of my recommendation. I also look forward to the rebuttal feedback and further discussions, and would be glad to raise my rating if thoughtful responses and improvements are provided.

---

## **------------------- Post-Rebuttal Summary --------------------**

The authors have not provided any response to the concerns and suggestions raised in my initial review during the rebuttal stage. The lack of engagement makes it difficult to assess whether or how the authors might address these fundamental concerns. Given that no clarification or improvement has been offered, I maintain my original comment that this submission falls below the acceptance threshold for ICLR.

---

### Comment · Reviewer_nywT · 2024-11-30
**Suggestions from Reviewer nywT to Submission 6406**

Dear Authors,

As the November 27th manuscript revision deadline has passed and we are now in the discussion period until December 3rd, I feel it important to provide my current evaluation and suggestions for Submission 6406.

This work presents a fundamental exploration of ConvNet architectures for vision tasks, yet the key concerns outlined in my detailed review regarding technical originality and theoretical foundations remain unaddressed. The technical points raised by myself and fellow reviewers have not yet received responses, leaving several critical questions unresolved.

I strongly encourage the authors to address these technical points with your responses during the remaining discussion period, which could provide valuable insights for strengthening this work, whether for the current stage or future submissions. I am also confident that the collaborative discussion process with reviewers could help further refine this research to meet the high standards expected in the community.

Given the current stage of the review process, while I maintain my rating of 5, I remain highly engaged and look forward to any responses or clarifications the authors may provide.

Best regards,

Reviewer nywT

---

### Comment · Reviewer_nywT · 2024-12-03
**Official Comments by Reviewer nywT**

Dear Authors and ACs,

As the reviewer designated for this submission, I note that the authors have not provided any response to the concerns and suggestions raised in my initial review during the author-reviewer discussion period. While the original submission presented some insights, several critical issues remain unaddressed.

The lack of engagement makes it difficult to assess whether or how the authors might address these fundamental concerns. Given that no clarification or improvement has been offered, I maintain my original rating of 5, indicating that this submission falls below the acceptance threshold for ICLR.

This conclusion is not a judgment on the potential merits but rather reflects the current state of the manuscript and the missed opportunity to address reviewers’ concerns in the rebuttal stage.

Best regards,

Reviewer nywT

---

### Meta-Review · Area_Chair_HEJa · 2024-12-18

**Metareview:**

The paper proposes a Simple and Fast Convolutional Neural Network (SFCNN) for vision tasks, which uses stacked 3×3 convolutions and a novel Global Sigmoid Linear Unit (GSiLU) activation function to achieve state-of-the-art performance while maintaining computational efficiency.

All reviewers have provided consistently negative ratings and the authors did not provide a response to address these issues. The final consensus of negative ratings lead to a rejection for this submission.

**Additional Comments On Reviewer Discussion:**

The authors did not provide a response to address the original concerns from reviewers. All reviewers remain the original ratings.

---

### Decision · Program_Chairs · 2025-01-22

Reject